# Hematological and Serum Biochemistry Values in Free-Ranging Crested Porcupine

**DOI:** 10.3390/vetsci7040171

**Published:** 2020-11-09

**Authors:** Francesca Coppola, Enrico D’Addio, Lucia Casini, Simona Sagona, Marco Aloisi, Antonio Felicioli

**Affiliations:** 1Departement of Veterinary Sciences, University of Pisa, Viale delle Piagge 2, 56124 Pisa, Italy; francesca.coppola@vet.unipi.it (F.C.); lucia.casini@unipi.it (L.C.); simona.sagona@unipi.it (S.S.); 2Freelance Practising Veterinary Surgeon, Serravezza, 55045 Lucca, Italy; dadvet74@gmail.com; 3Department of Pharmacy, University of Pisa, Via Bonanno 6, 56126 Pisa, Italy; 4Wildlife Rescue Center “CRASM Semproniano”, Loc. Casaccia snc, Semproniano, 58055 Grosseto, Italy; marcoaloisi@yahoo.it

**Keywords:** hematology, serum chemistry, health status, porcupine, *Hystrix cristata*, rodent

## Abstract

The crested porcupine is a widespread naturalized Italian rodent of African origin; nevertheless, very little information on the population abundance and its health status is available. In this study, the hematological and serum chemistry profile of 10 free-ranging captured crested porcupines was established for the first time. The mean hematological values resulted: 5.7 SD 0.4 M/μL for red blood cells; 13.6 SD 0.8 g/dL for hemoglobin; 77.3 SD 5.7 fL for mean corpuscular volume and 30.1 SD 4.7 g/dL for mean corpuscular hemoglobin concentration; 14.4 SD 7.2 K/μL for white blood cells; and 557.0 SD 469.9 K/μL for platelets. The mean urea and creatinine values resulted with 19.8 SD 8.3 mg/dL and 1.6 SD 3.0 mg/dL, respectively. The mean value of total protein was 6.7 SD 1.0 g/dL, with values of albumin higher than globulins. The mean activity of creatine kinase, aspartate transaminase, gamma-glutamyl transpeptidase, and alkaline phosphatase was 927.3 SD 607.6 U/L, 199.2 SD 70.8 U/L, 16.9 SD 13.7 U/L, and 256 SD 75.8 U/L, respectively. Highest values of alkaline phosphatase were recorded in two porcupines presenting severe injuries with clear signs of infection. These preliminary results may be a helpful tool in order to assess porcupine health status.

## 1. Introduction

The crested porcupine (*Hystrix cristata*) is a naturalized Italian rodent of African origin widely spread throughout the Italian territory [1,2]. It is a monogamous, burrowing, mainly herbivorous and nocturnal mammal [3,4,5,6,7,8]. The crested porcupine is strictly protected by European and Italian Law since 1977 and it is included in the International Union for Conservation of Nature (IUCN) Red List of threatened species among the “Least concern” species. Habitat diversity and environmental fragmentation are a key factor for the presence of this rodent [9]. However, the crested porcupine is able to adapt itself to different habitat types where vegetation offers adequate woody and/or bush covered areas and food availability [9]. In the Italian agro-ecosystem, the crested porcupine has benefitted from the increase in the production of agricultural crops, leading to an increase of porcupine–human interaction and conflicts [10,11].

The crested porcupine is considered a pest species throughout all its distribution range [11]. Trophic activity of the crested porcupine in central Italy causes damage to crops in cultivated areas due to animal preference for maize, potatoes, sunflowers, pumpkin, and melon [9,12] and it is also reported to be responsible for trees debarking [9]. Crested porcupine crop damages occur mainly in private vegetable gardens in the surroundings of human settlements [11]. Porcupine–human conflicts are also locally increased due to damage performed to manmade riverbanks in which this rodent digs its burrows with consequent collapse of the banks themselves, causing flood events [13]. Therefore, for these reasons and also for its tasty meat, the poaching of this rodent is a widespread practice in Italy [14].

Despite the strict protectionist status and the high conservation interest of this rodent, in Italy, there are no data concerning the abundance of the porcupine population and very little knowledge is available on its health status. Recent studies highlighted that the crested porcupine can be infected by *Giardia duodenalis* [15], several *Leptospira* serogroups and is considered a new host for *Leptospira* serovar Pomona [16,17]. This evidence suggests that this rodent could be a new potential source in the epidemiology routes of zoonotic diseases such as *Giardiasis* and *Leptospirosis*. In this context, occasional settlements sharing and cohabitation between porcupines, badgers, and red foxes [18] and the scavenging behavior in porcupines [8] seem to play a key role in the transmission of such diseases.

Hospitalization of either injured or ill individuals and/or porcupettes in wildlife rescue centers (WRC) has undergone a strong increase in the last decades. However, despite WRC having a key role in wildlife conservation, in most cases, porcupines die during hospitalization due to the severity of injuries or difficulty in making a diagnosis [19]. Among hystricomorph rodents, hematological and biochemical parameters were investigated in coypu (*Myocastor coypus*) [20], capybara (*Hydrochoerus hydrochaeris*) [21,22], and guinea pig (*Cavia porcellus*) [23]. Conversely, concerning porcupines, only few data are available for three species of New World porcupines: captive bristle-spined porcupine (*Chaetomys subspinosus*) [24], Brazilian porcupine (*Coendou prehensilis*), and black-tailed hairy dwarf porcupine (*Coendou melanurus*) [25]. The knowledge of the hematological and blood chemistry profile could be a useful diagnostic and prognostic tool; nevertheless, no data are yet available on these parameters for Old World porcupines and among which is the crested porcupine.

In this study, the hematological and serum chemistry profile of free-ranging captured crested porcupines were preliminary established for the first time during a wider investigation on the porcupine health status.

## 2. Matherials and Methods

### 2.1. Samples Collection

Porcupines capture campaigns were performed between July 2019 and August 2020 both in a hilly area in Crespina-Lorenzana (43.57181 Lat.–10.55348 Long.) and in the wildlife hunting reserve Camugliano (43.60210 Lat—10.64742 Long.) in the Province of Pisa (Tuscany, Central Italy). The capture-marking activity and handling protocols of porcupines was approved by the Italian Institute for Environmental Protection and Research (ISPRA) with protocol number 22,584 of the 8 May 2017 and protocol number 150,071 of the 16 March 2018 and by Tuscany Region with the Decree n. 14,235 of the 3 October 2017 n. 4842 of the 6 April 2018. In each area, six capture traps were placed along pathways connecting porcupine settlements to food-patch areas, where porcupine presence signs (i.e., quills, footprints, and feces) were regularly detected. The traps were checked two times/day (i.e., 8:00 am and 7:00 pm) and the non-target species captured were immediately released. The porcupines were trapped in wire mesh cages with a double entrance (110 × 42 × 42 cm), baited with corn, and monitored by camera-traps. Camera-traps were set to record 30 s long video clips, with hour and date displayed, and without time-lapse. Each capture porcupine was weighed, anesthetized, sexed, and the age class was estimated based on animal weight [7]. Moreover, individual marks with color adhesive tapes on the quills or white/black paint sprayed on the body were applied to each captured porcupine in order to recognize it in case of recapture. Porcupines were anesthetized with Zoletil 100^®^ by intramuscular injections in the lumbar region using an air-compressed syringe (Mini-ject 2000, 2 mL) administered by a blow-pipe 1 m long. The injection dose was 5 mg/kg. Each captured porcupine was subject to a clinical examination (visual inspection of teeth, skin, and all mucous membranes, palpation of the abdomen, and chest auscultation) and the body temperature and heart rate were also recorded.

A 5–10 mL sample of blood was collected from the brachiocephalic vein of each animal using a butterfly needle (21 G × 3/4″, 0.8 × 19 mm) attached to a sterile 5 mL syringe. Each blood sample was divided in two aliquots, one in 4 mL tubes with serum separating granules without clot activator and the other in K3-EDTA tubes (Aptaca, Asti, Italy). Blood samples were kept in a cooler bag after field collection and stored at 4 °C until analysis.

### 2.2. Hematology and Serum Chemistry Analysis

All blood samples were analyzed on the same day of collection. Blood samples containing clots or grossly hemolyzed at visual inspection after centrifugation were not processed. The samples with EDTA were analyzed for a complete blood cell count by using an auto hematology analyzer (BC-2800 Vet, Mindray, Milano, Italy) which uses the electrical impedance method for counting and the cyanide free method for hemoglobin. The hematological values analyzed included: red blood cells (RBC), hemoglobin (Hgb), mean corpuscular volume (MCV), mean corpuscular hemoglobin (MCH), mean corpuscular hemoglobin concentration (MCHC), red blood cells distribution width (RDW), platelets (PLT), mean platelet volume (MPV), platelet distribution width (PDW), plateletcrit (PCT), white blood cells (WBC), lymphocytes, monocytes, granulocytes, and eosinophils.

Blood samples without EDTA were centrifuged at 3000 rpm for 10 min to collect serum from blood. Serum chemistry parameters, including total proteins, albumin, globulins, glucose, total bilirubin, creatinine, urea, gamma-glutamyl transpeptidase (GGT), alkaline phosphatase (ALP), aspartate transaminase (AST), alanine aminotransferase (ALT), creatine kinase (CK), lactate dehydrogenase (LDH), sodium (Na^+^), potassium (K^+^), Na^+^/K^+^ ratio, calcium (Ca^2+^), phosphates, cholesterol, chloride, and osmolality were measured using an automated chemistry analyzer (Catalyst One, IDEXX, Westbrook, ME, USA) which uses dry slide technology. Descriptive data were presented as mean, standards deviation, minimum and maximum value, and median value (JMP, SAS Institute, 2008).

## 3. Results

Overall, a total of 15 porcupines were captured. Blood samples for hematological and biochemical analyses were collected from 13 out of 15 captured porcupines: 6 adult females, 2 adult males, 3 sub-adult females, 1 sub-adult male, and 1 female porcupette. At clinical examination, 11 out of 13 captured porcupines appeared in good physical condition while only 2 porcupines presented several body injuries. One captured adult female was lactating (i.e., milk ejected from the turgid mammary glands at palpation) and another captured adult female was pregnant. On the 13 blood samples collected, only 3 were not processed because they contained clots or were grossly hemolyzed. Serum chemistry parameters were obtained from all processed samples while hematological values were obtained from 7 out of 10 (70%) collected samples.

In Table 1 are reported the age class, sex, body weight, body temperature, heart rate, and the clinical status of each captured porcupines of which blood samples were collected and analyzed. In Table 2 and Table 3, the biochemical and hematological profile obtained are reported.

The mean values of RBC resulted 5.7 SD 0.4 M/μL with an MCV of 77.3 SD 5.7 fL, while the mean values of Hgb and MCHC resulted with 13.6 SD 0.8 g/dL and 30.1 SD 4.7 g/dL, respectively. The mean value of WBC was 14.4 SD 7.2 K/μL and the PLT of 557.0 SD 469.9 K/μL. Highest values of WBC and PLT were record in two captured porcupines in which severe injuries and clear signs of infection were detected.

The mean value of total protein resulted with 6.7 SD 1.0 g/dL with recorded values of albumin (2.8 SD 0.2 g/dL) lower than globulins (3.5 SD 0.3 g/dL). Mean values of urea and creatinine in the crested porcupine resulted with 19.8 SD 8.3 mg/dL and 1.6 SD 3.0 mg/dL, respectively. Only in one adult male, the value of creatinine of 10.1 mg/dL associated with a high value of urea (39 mg/dL) was recorded. The mean value of GGT was 16.9 SD 13.7 U/L. In the 66.7% (*n* = 9) of samples, the GGT value ranged between 7 to 9 U/L, while in 3 out of 9 (33.3%) samples, values higher than 30 U/L were recorded. The CK and AST activity ranged from 236–2238 U/L and 100–265 U/L, respectively, with mean values of CK recorded of 927.3 SD 607.6 U/L and AST of 199.2 SD 70.8 U/L. The mean values for ALP was 256 SD 75.8 U/L with highest values of ALP recorded in the two individuals presenting severe injuries with clear signs of infection.

## 4. Discussion

The preliminary biochemical and hematological profile of the free-ranging crested porcupine was obtained here for the first time. To the best of our knowledge, hematologic parameters and serum chemistry in the Old World porcupine has been previously investigated only by Leonetti [26] on a small sample (*n* = 6) of hospitalized crested porcupines. The same author also reported the hematological profile of one healthy porcupine and some biochemical values (i.e., urea, creatinine, total bilirubin, GGT, AST, and CK) of two free-ranging individuals [26]. Healthy porcupines analyzed by Leonetti [26] were captured in the same areas of this study by some authors of this paper as part of a previous investigation on crested porcupine biology. All the hematological and serum chemistry values obtained in this investigation were in accordance with those reported by Leonetti [26] on healthy porcupines. The mean values of red blood cells (5.7 SD 0.4 M/μL) in the crested porcupine resulted higher than those reported in some species of new world porcupines by De Almeida et al. [24] (3.45 SD 0.45 M/μL) and Moreau et al. [25] (3.6 SD 0.5 M/μL and 3.5 SD 0.4 M/μL). The RBC values obtained in the crested porcupine were also higher than those reported in capybara, whose RBC values range from 1.5 to 3.88 M/μL (2.78 SD 0.38 M/μL) [21,22]. The RBC values of the crested porcupine are within the range obtained for coypu, (3.9 to 6 M/μL; 4.5 SD 3.6 M/μL) [20] and guinea pig (5.0–5.7 in female and 5.7–6.4 in male) [23]. These last species are the only wild (capybara and coypu) and domestic (guinea pig) rodents belonging to the same suborder of porcupines (Hystricomorpha) but not to the same family.

In the crested porcupine, lower MCV values (77.3 SD 5.75 fL) were recorded when compared with the captive bristle-spined porcupine (96.1 SD 3.2 fL) [24], Brazilian porcupine (94 SD 6 fL), black-tailed hairy dwarf porcupine (93.7 SD 5.4 fL) [25], capybara (132.27 SD 10.3 fL), and coypu (96.8 SD 2.1 fL) [20,22]. Conversely, MCV values in the crested porcupine resulted in accordance with those reported in guinea pig (IR = 74–85 in female and 71–82 in male) [23]. Furthermore, the crested porcupine showed similar values of hemoglobin (13.6 SD 0.8 g/dL) and MCHC (30.1 SD 4.8 g/dL) than captive bristle-spined porcupine, Brazilian porcupine, black-tailed hairy dwarf porcupine, capybara, coypus, and guinea pig [20,22,23,24,25]. These results seem to be in accordance with the negative correlation between RBC size and RBC number with invariant values of MCHC and hemoglobin previously observed in different vertebrate group [27].

The mean white blood cells (WBC) and platelets (PLT) values in the crested porcupine were 14.4 SD 7.2 K/μL and 557.0 SD 469.9 K/μL, respectively. These values result higher than those reported in the captive bristle-spined porcupine (WBC = 8.35 SD 1.88 K/μL, PLT = 476.5 SD 41.3 K/μL), Brazilian porcupine (WBC = 7.9 SD 3.8 K/μL, PLT = 278 SD 117 K/μL), black-tailed hairy dwarf porcupine (WBC = 8.4 SD 4.8 K/μL, PLT = 370 SD 194 K/μL), capybara (WBC = 6.75 SD 2.9 K/μL), and guinea pig (WBC = 5.3–7.3 K/μL; PLT = 201–304 K/μL) [22,23,24,25]. Mean value of WBC in the crested porcupine was also higher than in the coypu (11.0 SD 1.8 K/μL), while the mean PLT value fit within the reference interval was recorded in the coypu (534–727 K/μL) [20]. However, the highest values of WBC and PLT in the crested porcupine were recorded only in two captured individuals in which severe injuries and clear signs of infection were detected. Therefore, the crested porcupine WBC (9.3 K/μL) and PLT (454 K/μL) median values obtained are to be considered the most reliable.

The mean value of total protein obtained in the crested porcupine resulted with 6.7 SD 1.0 g/dL with values of albumin (2.8 SD 0.2 g/dL) lower than globulins (3.5 SD 0.3 g/dL) and the albumin/globulins rate ranging from 0.7 to 0.9. No data are available on the albumin/globulins rate in the Brazilian porcupine, black-tailed hairy dwarf porcupine, capybara, coypu, and guinea pig. However, TP and albumin values obtained in the crested porcupine are in accordance with those reported for capybara (TP = 6.7 SD 0.49 g/dL, albumin = 2.36 SD 0.25 g/dL) [22], Brazilian porcupine (TP = 6.7 SD 0.9 g/dL), and black-tailed hairy dwarf porcupine (TP = 6.8 SD 0.8 g/dL) [25]. Moreover, the value of TP in the crested porcupine falls in the reference range recorded for coypu (6.3 to 8.9 g/dL) [20] while resulted higher than those reported in guinea pig (5.1 to 5.7 g/dL) [23]. On the other hand, the crested porcupine values of albumin were lower than those obtained in coypu (3.0 to 6.1 g/dL) [20] while fit within the range reported in guinea pig (2.7 to 3 g/dL) [23]. Albumin is the primary and most homogenous fraction; comprising 35% to 50% of total serum proteins in animals [28] while globulins are divided in four fractions: α1, α2, β, and γ and the latest including the immunoglobulins IgA, IgM, IgE, and IgG [29]. In this investigation, albumin values were obtained by colorimetric Bromocresol green assay and total globulins value were obtained by difference between total protein and albumin. Increase in the globulin fraction usually results from an increase in immunoglobulins, but in pathologic states, increase of other fractions with different electrophoretic patterns can occur [29]. Therefore, electrophoresis analysis on serum protein fractions in the crested porcupine is desirable and could be a useful diagnostic and prognostic tool for health monitoring of this rodent as previously reported in other birds and mammals including humans [28,30,31].

Average values of urea and creatinine in the crested porcupine resulted with 19.8 SD 8.3 mg/dL and 1.6 SD 3.0 mg/dL, respectively. Mean value of urea recorded in the crested porcupine was in accordance with those reported in the free-ranging black-tailed hairy dwarf porcupine (20.55 SD 16.2 mg/dL) [25], falling in the range obtained in the free-ranging coypu (11.3 to 36.8 mg/dL) [20] and guinea pig (18 to 23 mg/dL) [23], while it resulted higher than the free-ranging Brazilian porcupine (10.6 SD 7.8 mg/dL) and lower than the free-ranging capybara (26.3 to 70.2 mg/dL) [22,25]. On the other hand, the creatinine mean value obtained from the crested porcupine fits within the range recorded in the coypu (0.5 to 5.8 mg/dL) [20] and was in accordance with those obtained in the capybara (0.6 to 1.5 mg/dL) [22]. Conversely, crested porcupine creatinine values were lower than in the black-tailed hairy dwarf porcupine (4.75 SD 0.11 mg/dL) and Brazilian porcupine (5.75 SD 0.1 mg/dL) [25] and higher than the guinea pig (0.1 to 0.6 mg/dL) [23]. Therefore, values of urea and creatinine obtained seem to indicate a good health status of captured crested porcupines. Only in one adult male porcupine, the value of creatinine of 10.1 mg/dL associated with a high value of urea (39 mg/dL) was recorded. Serum creatinine and urea concentrations are usually used as indicators for kidney disease diagnosis in other mammal species [32]. A study recently performed on the health status of crested porcupines highlighted a high seroprevalence of *Leptospira* in this rodent [16]. Moreover, isolation of *Leptospira* serovar Pomona has been also recently performed in a porcupette [17]. Therefore, a possible hypothesis is that the high values of urea and creatinine recorded could be an indicator of kidney damage due to *Leptospira* infection.

Among the investigated serum enzymes, the mean value of GGT activity (16.9 SD 13.7 U/L) recorded in the crested porcupine resulted higher than values reported in the capybara (1.3 to 4.5 U/L) [22], Brazilian porcupine (8 SD 2 U/L), black-tailed hairy dwarf porcupine (5 SD 3 U/L) [25], and guinea pig (6 to 13 U/L) [23]. However, in three porcupines blood samples, GGT values resulted higher than 30 U/L, probably due to the presence of unknown pathological states. For this reason, the GGT median value obtained of 9 U/L is probably the most reliable and is in accordance to those reported in the Brazilian porcupine and black-tailed hairy dwarf porcupine and guinea pig [23,25]. Activity of CK and AST in captured crested porcupines ranged from 236 to 2238 U/L and from 100 to 265 U/L, respectively, with highest values obtained in those porcupines whose permanence in the cage traps exceeded 10 h (i.e., captured from 8:00–9:00 pm and anesthetized from 8:00–10:00 am). LDH activity in the crested porcupine was recorded only in one blood sample (2727 U/L) thereby not comparable with LDH activity in other species. Activity of CK and AST in the crested porcupine resulted higher than those reported in the wild black-tailed hairy dwarf porcupine (CK = 541 SD 680 U/L; AST = 146 SD 56 U/L), Brazilian porcupine (CK = 646 SD 862 U/L; AST = 133 SD 49 U/L) [25], in free-ranging coypus (CK = 182–552 U/L; AST = 118–177 U/L), and guinea pig (AST = 35–54 U/L) [20,23]. These results are probably due to muscle or soft-tissue injury following excitement and stress during capture which may increase the serum CK and AST activity [25]. In this study, AST activity of the blood serum in the crested porcupine resulted higher than ALT activity as previously reported in the coypu and guinea pig [20,23].

The ALP activity (256 SD 75.8 U/L) recorded in crested porcupines fits with those determined for the coypu (200 to 399 U/L) [20] while resulted higher than in the Brazilian porcupine (120 SD 48 U/L), black-tailed hairy dwarf porcupine (166 SD 150 U/L) [25], and guinea pig (32 to 67 U/L) [23]. In domestic animals, four ALP variants: intestinal ALP, liver ALP, bone-specific ALP (BALP), and corticosteroid-induced ALP (CALP) have been identified, which renders difficult the interpretation of total ALP values [33].

However, total ALP activity can be a useful indicator for skeletal development or skeletal diseases as well as of disease processes [33]. Moreover, neutrophil alkaline phosphatase (NAP) is also present in the secretory vesicles or on the plasma membrane of neutrophils. The increase of NAP is one of the main responses of neutrophils to bacterial infections both in humans and animals [34,35,36]. In the crested porcupine, the highest values of ALP were recorded in two individuals presenting severe injuries with clear signs of infection while no differences were recorded in ALP values between sub-adult and adult individuals. Therefore, the higher values of ALP activity recorded in this study are probably due to bacterial infections. As for ALP, care should be also taken when interpreting other values such as CK and AST since high values mainly resulting from stress conditions.

Among serum chemistry values in the crested porcupine, sodium (Na^+^), potassium (K^+^), Na^+^/K^+^ ratio, phosphates, chloride, and osmolality were also obtained but only from a single sample and thereby not comparable. However, as there are no other reference values for these parameters in the crested porcupine, we believe noteworthy to provide these data.

## 5. Conclusions

In conclusion, the authors are aware of the small sample size, nevertheless few data are available concerning these parameters in Old World porcupines, and in particular, on crested porcupines as well as on health status of this species. Therefore, since crested porcupine hospitalization in wildlife rescue centers is strongly increasing, the results obtained in this study may be a helpful prognostic and diagnostic tool in order to assess the health status of this rodent in the wild. These preliminary results may also be a useful knowledge base for the development of hematological and serum biochemistry reference values. Moreover, since seasonal variations and differences between sexes of blood parameters have been recorded in other rodent species, further investigations on seasonal changes in hematological and serum biochemistry values and sex differences in the crested porcupine are desirable.

## Figures and Tables

**Table 1 vetsci-07-00171-t001:** Age class, sex, body weight (BW), clinical status, body temperature (BT), and heart rate (HR) of each captured porcupine from which blood samples were collected and analyzed.

Age Class	Sex	BW (kg)	Clinical Status	BT (°C)	HR (bpm)
Adult	Male	11.9	Severe abscess in the rump	36	110
Adult	Female	14.7	Good, Pregnant	38.2	129
Sub-adult	Male	10.6	Good	38.2	113
Sub-adult	Female	9.4	Left ear laceration	37.2	112
Adult	Female	15.2	Good, Lactation	38.5	116
Adult	Female	11	Good	38.1	119
Sub-adult	Female	5	Good	39	121
Adult	Male	14.5	Good	38.2	116
Adult	Female	11	Good	38.6	111
Sub-adult	Female	9.5	Good	37.9	114

**Table 2 vetsci-07-00171-t002:** Biochemical values of crested porcupine blood. For each parameter, the mean, the standard deviation (SD), minimum (Min) and maximum (Max) value, and the median are reported.

Parameter	*n*	Mean	SD	Min	Max	Median
Glucose (mg/dL)	6	179	31.2	154	231	169
Creatinine (mg/dL)	10	1.6	3.0	0.3	10.1	0.7
Urea-N BUN (mg/dL)	10	19.8	8.3	10	39	19
BUN/Creat	5	41.2	21.1	16	68	41
TP (g/dL)	7	6.7	1.0	6	9.1	6.3
Albumin (g/dL)	5	2.8	0.2	2.5	2.9	2.8
Globulins (g/dL)	5	3.5	0.3	3.3	4	3.5
Alb/Glob	5	0.8	0.1	0.7	0.9	0.8
GGT (U/L)	9	16.9	13.7	7	32	9
ALP (U/L)	5	256	75.8	195	349	205
AST (U/L)	6	199.2	70.8	100	265	212.5
ALT (U/L)	6	25.3	2.5	23	30	24.5
CK (U/L)	8	927.3	607.6	236	2238	841.5
LDH (U/L)	1	2727	-	2727	2727	2727
Na^+^ (mmol/L)	1	145	-	145	145	145
K^+^ (mmol/L)	1	4.8	-	4.8	4.8	4.8
Ca^2+^ (mg/dL)	2	9.7	0.1	9.6	9.7	9.65
Na/k	1	30	-	30	30	30
Phosphates (mg/dL)	1	6.2	-	6.2	6.2	6.2
Chloride (mmol/L)	1	104	-	104	104	104
Bil Tot (mg/dL)	6	0.7	0.6	0.1	1.7	0.5
Cholesterol (mg/dL)	2	25	26.9	6	44	25
Osmolality (mmol/kg)	1	293	-	293	293	293

TP = total proteins, GGT = gamma-glutamyl transpeptidase, ALP = alkaline phosphatase, AST = aspartate transaminase, ALT = alanine aminotransferase, CK = creatine kinase, Na^+^ = sodium, K^+^ = potassium, Ca^2+^ = calcium, Bil Tot = total bilirubin, LDH = lactate dehydrogenase.

**Table 3 vetsci-07-00171-t003:** Hematological profile of the crested porcupine. For each parameter, the mean, the standard deviation (SD) minimum (Min) and maximum (Max) value, and the median are reported.

Parameter	*n*	Mean	SD	Min	Max	Median
RBC (M/μL)	7	5.7	0.37	5.3	6.25	5.7
Hgb (g/dL)	7	13.6	0.8	12.7	14.6	13.2
Hct (%)	7	43.8	4.7	36	49.7	44.4
MCV (fL)	7	77.3	5.7	68	82.3	79.6
MCH (pg)	7	23.8	0.9	22.7	24.9	23.5
MCHC (g/dL)	7	30.1	4.77	22.7	36.6	29.3
RDW (%)	5	16.9	1.33	15.4	18.9	16.7
PLT (K/μL)	7	557.0	470.5	86	1479	454
MPV (fL)	5	6.0	1.36	4.5	8	5.8
PDW	6	17.5	1.15	15.8	18.5	17.8
PCT (%)	5	0.3	0.17	0.146	0.562	0.216
WBC (K/μL)	7	14.4	7.2	8.1	22.5	9.3
Lymphocytes (K/μL)	7	4.3	3.56	1.1	10	2.8
Lymphocytes (%)	7	26.8	11.4	12.9	44.6	26.1
Monocytes (K/μL)	7	0.5	0.2	0.3	0.8	0.6
Monocytes (%)	7	3.3	0.6	2.7	3.9	3.7
Granulocytes (K/μL)	7	9.5	4.20	5.5	15.8	7.5
Granulocytes (%)	7	69.7	10.85	56.2	83.3	72.6
Eosinophils (%)	3	1.5	0.12	1.4	1.6	1.4

RBC = red blood cells, Hgb = hemoglobin, Hct = hematocrit, MCV = Mean Cell Volume, MCH = mean corpuscular hemoglobin, MCHC = mean corpuscular hemoglobin concentration, RDW = red blood cells distribution width, PLT = platelets, MPV = mean platelet volume, PDW = platelet distribution width, PCT = plateletcrit, WBC = white blood cell.

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
