# Peer review of "Hematological and Serum Biochemistry Values in Free-Ranging Crested Porcupine"

_vetsci, 2020, doi:10.3390/vetsci7040171_

Round 1

Reviewer 1 Report

Very interesting and appropriate work also considering typology of the species and scientific address chosen on the territory.

Introduction , Matherials and Methods  included samples collection are adequated , results interesting and innovative also in terms of innovation and biodiversity conservation, discussion is wide , well described but very long and articulate.

There is no stated conclusions and it is a pity.

Therefore it will require to the Authors to enter, please,   your conclusions , by unplugging them them from the discussion. 

Reviewer 2 Report

The manuscript presents haematology and blood chemistry for a rodent. The blood data provided in the paper will provide health measures for the crested porcupine. Care needs to be taken with using common and scientific names, the paper switches between the two throughout for a number of species. I would suggest using the convention of: common name with scientific name in brackets on the first mention and then using the common name throughout the rest of the manuscript for all animals. Some things that are not considered in this paper are seasonal changes in blood parameters levels and sex differences. I understand these were outside of the scope of your study however they could be mentioned in the discussion section and here you could also suggest a future study to develop reference values incorporating season and sex as factors to investigate.

Specific points:

L22 – the value for GGT does not match that presented in the Table
L35 – change bushes to bush

L42 – change humans’ to singular human

L76 – add :00 after 7pm to be consistent with 8:00 am

L76 – change no-target to non-target

L116 – change breastfeeding to lactating

L121 – change hart to heart

L130 –previously you used the common name for porcupine why have you gone to scientific name here?

L154 – change “At” to “To”

L162 & L171 – are examples of where you have changed between scientific and common names

L221 – change creatinine to creatinine

L228 – Give leptospira a capital L and italicise

L234 – “For this reasons” change to “For this reason” or “For these reasons”

L265 – you should also mention here that care should be taken when interpreting some values such as CK as they are likely elevated from stress

Table 2 – there are two places that have comas that should be changed to decimal places

Reviewer 3 Report

Overall, this paper provides valuable information to further this field of research.  It would have been helpful to include plasma protein electrophoresis data to provide baseline data for some of the plasma proteins discussed. More details about blood collection is also necessary.

Line 80:  ‘weighed’ instead of ‘weighted’

Line 80 and 83: two different spellings of ‘anesthetised’ and ‘anaesthetized’ were used; please reconcile

Line 80:  ‘on the base’ should be ‘based on animal weight’

Line 82: ‘porcupines’ should be ‘porcupine’

Line 89:  What gauge and length of needle was used for blood collection?  What type of tubes were used for collection – were they Vacutainer or another type? Were they immediately transferred to a cooler at 4 C or were they kept at a different temperature in the field?

Line 94: what criteria was used to determine ‘grossly haemolysed’?

Line 102: ‘3.000’ should be ‘3,000’

Line 108: ‘use’ should be ‘uses’

Lines 121-123:  please capitalize also instance of Table when it is followed by a table number

Line 121:  ‘hart’ should be ‘heart’

Line 140: ‘porcupines’ should be ‘porcupine’

Line 140: ‘of which blood sample’ should be ‘from which blood samples’

Table 1: ‘Sever’ should be ‘Severe’

Line 153: ‘here obtained’ should be ‘obtained here’

Line 183: C. Subspinosus should be lower case ‘s’

Line 183: Prehensilis should be lower case ‘p’

Line 184: Melanurus should be lower case ‘m’

Line 186: Coypus should be lower case ‘c’

Paragraph starting on line 191:  Why are plasma electrophoresis data discussed when these analyses were not conducted ?  If they were conducted, please provide a representative image of an electrophoretogram.

Line 193:  species should all be lower case

Line 200: most instances of ‘fitted’ as used in this section should be changed to ‘fit’

Line 205: ‘result’ should be ‘results’

Line 221: ‘creatinina’ should be ‘creatinine’

Line 224: ‘diseases’ should be ‘disease’

Line 254: ‘neutrophils’ should be ‘neutrophil’ in the term ‘neutrophil alkaline phosphatase’
